# Gene Expression of Ethanol and Acetate Metabolic Pathways in the *Acinetobacter baumannii* EmaSR Regulon

**DOI:** 10.3390/microorganisms12020331

**Published:** 2024-02-04

**Authors:** Yu-Weng Huang, Hung-Yu Shu, Guang-Huey Lin

**Affiliations:** 1Department of Biomedical Sciences and Engineering, School of Medicine, Tzu Chi University, Hualien 970374, Taiwan; 110726102@gms.tcu.edu.tw; 2Department of Bioscience Technology, Chang Jung Christian University, Tainan 711301, Taiwan; hyshu@mail.cjcu.edu.tw; 3Master Program in Biomedical Sciences, School of Medicine, Tzu Chi University, Hualien 970374, Taiwan; 4International College, Tzu Chi University, Hualien 970374, Taiwan

**Keywords:** two-component system, ethanol metabolism, acetate metabolism, *Acinetobacter baumannii*, acetate: succinyl-CoA transferase (ASCT)

## Abstract

Background: Previous studies have confirmed the involvement of EmaSR (ethanol metabolism a sensor/regulator) in the regulation of *Acinetobacter baumannii* ATCC 19606 ethanol and acetate metabolism. RNA-seq analysis further revealed that *DJ41_568-571*, *DJ41_2796*, *DJ41_3218*, and *DJ41_3568* regulatory gene clusters potentially participate in ethanol and acetate metabolism under the control of EmaSR. Methods: This study fused the EmaSR regulon promoter segments with reporter genes and used fluorescence expression levels to determine whether EmaSR influences regulon expression in ethanol or acetate salt environments. The enzymatic function and kinetics of significantly regulated regulons were also studied. Results: The EmaSR regulons P*_2796_* and P*_3218_* exhibited > 2-fold increase in fluorescence expression in wild type compared to mutant strains in both ethanol and acetate environments, and P*_emaR_* demonstrated a comparable trend. Moreover, increases in DJ41_2796 concentration enhanced the conversion of acetate and succinyl-CoA into acetyl-CoA and succinate, suggesting that DJ41_2796 possesses acetate: succinyl-CoA transferase (ASCT) activity. The k_cat_/K_M_ values for DJ41_2796 with potassium acetate, sodium acetate, and succinyl-CoA were 0.2131, 0.4547, and 20.4623 mM^−1^s^−1^, respectively. Conclusions: In *A. baumannii*, EmaSR controls genes involved in ethanol and acetate metabolism, and the EmaSR regulon DJ41_2796 was found to possess ASCT activity.

## 1. Introduction

Ethanol is a widely-used disinfectant, but previous research has shown that sub-inhibitory concentrations of ethanol are not only unable to eradicate bacteria but can conversely promote biofilm formation to increase bacterial resistance [1,2,3,4]. Ethanol concentrations of 2.3–3.5% enhance the ability of *Staphylococcus aureus* to form biofilms, while 1–2% concentrations are sufficient to induce biofilm formation in *Pseudomonas aeruginosa* [1,2,3,4]. The presence of alcohol dehydrogenases (Adhs) can further help bacteria to degrade alcohols, reducing ethanol-induced damage and transforming ethanol into a carbon source for growth instead [5,6].

Acetate is a type of short-chain fatty acid (SCFA) produced during ethanol metabolism. Previous research has shown that SCFAs inhibit toxin-producing bacteria such as *Bacillus cereus* and *Clostridium difficile*, as well as foodborne pathogens such as *Campylobacter* and *Salmonella* [7]. Additionally, it has been found that acetate increases the activity of macrophages to enhance their lethality against *Streptococcus pneumoniae* [8]. However, many types of bacteria can also metabolize acetate; for example, *Escherichia coli* can convert acetate into acetyl-CoA via acetate kinase (AckA) and phosphotransacetylase (Pta), or through acetyl-CoA synthetase [9]. Both of these pathways require the consumption of ATP, but in *Pseudomonas* sp., *Klebsiella pneumoniae*, and *Acetobacter*, it has been observed that acetate:succinyl-CoA transferase (ASCT) can transfer the CoA from succinyl-CoA to acetate to produce acetyl-CoA without consuming ATP [10].

Bacteria often use a two-component system (TCS) to detect and respond to changes in the environment. A typical two-component system consists of a histidine sensor kinase and a response regulator. Previous research has identified an ErdSR TCS that regulates ethanol metabolism [11,12]; in addition, MxtR/ErdR in *Pseudomonas putida* KT2440 was found to regulate the metabolism of acetate and pyruvate, partly through enzymes such as ScpC (PP_0154), which functions as an ASCT [13]. Recent research on *Acinetobacter baumannii*, *K. pneumoniae*, and *S. aureus* has also confirmed that many of their TCSs have regulatory functions in metabolism [14,15,16].

*A. baumannii* is a common pathogen in nosocomial infections. Previous research identified seven Adh genes in the genome of *A. baumannii* ATCC 19606, of which Adh exhibits a higher affinity for ethanol over other alcohols; the absence of Adh4 specifically hindered bacterial growth in low concentrations of ethanol, propanol, and butanol [17]. Subsequently, it was confirmed that the absence of the *emaSR* TCS in *A. baumannii* leads to an inability to grow in low concentrations of ethanol, as EmaSR was shown to regulate ethanol metabolism. Additionally, it was discovered that EmaSR shares approximately 40% or more amino acid sequence similarity with ErdSR and MxtR/ErdR, which regulate ethanol metabolism in *P. aeruginosa* PAO1. Furthermore, the loss of EmaSR was found to lead to growth defects in *A. baumannii* when exposed to a 20 mM acetate environment [18]. However, in low-concentration ethanol environments, transcriptome analysis revealed that EmaSR significantly upregulated several genes annotated for involvement in acetate and pyruvate metabolic pathways (Figure 1), including *DJ41_568-571*, *DJ41_2796*, *DJ41_3218*, and *DJ41_3568* [18]. Considering that the presence of ethanol, which is widely used in hospital environments as a disinfectant, has been reported to induce stress responses in *A. baumannii* that can lead to increased virulence [17], understanding ethanol and acetate metabolic pathways can be crucial to formulating new strategies aimed at preventing or mitigating nosocomial and opportunistic infections.

In this study, we investigated the expression of a reporter gene fused with the promoter of EmaSR regulons, in order to confirm the metabolic pathways through which EmaSR regulates bacterial metabolism of ethanol and acetate. We also conducted an enzymatic analysis to identify EmaSR regulons with crucial enzymatic activity in the *A. baumannii* metabolism of ethanol and acetate. Our results provide further insight into the role of the EmaSR TCS in *A. baumannii* resilience and may have implications for other pathogenic bacterial species as well.

## 2. Materials and Methods

### 2.1. Bacterial Strains, Plasmids, Primers, and Growth of Strains

All strains of *E. coli* and *A. baumannii* were grown in LB medium at 37 °C with shaking to increase the bacterial count; however, the fluorescence expression experiments were carried out with cultures using an M9 medium (composition described in Section 2.2). Based on the antibiotic resistance of each strain, the culture medium was supplemented with a final concentration of 50 µg/mL ampicillin, 50 µg/mL kanamycin, or 12.5 µg/mL tetracycline. Bacterial strains and plasmids are described in Table 1 and Table 2, respectively. Primers used for the construction and verification of the recombinant strains are presented in Table 3.

### 2.2. Green Florescence Assay

TCSG represents the abbreviation of the **t**wo-**c**omponent **s**ystem with ***g****fp* gene. This reporter system utilized the *Pst*I and *Xba*I restriction enzyme sites to combine the promoter fragment with *gfpuv*, which contains three amino acid modifications that significantly increase fluorescence emissions without affecting emission wavelength [22]. Each promoter-*gfpuv* fragment were then recombined into pWH1266 using Gibson assembly (E2611, NEB, Ipswich, MA, USA). The recombinant plasmids were introduced into *A. baumannii* strains by electroporation at 1.8 kV. The construction of each strain was confirmed by a colony PCR, and the analysis results are presented in Appendix A.

Strains with the pWH1266G reporter plasmid were cultured in an M9 medium (33.7 mM Na_2_PO_4_, 22 mM KH_2_PO_4_, 8.55 mM NaCl, 9.35 mM NH_4_Cl, 1 mM MgSO_4_, 0.3 mM CaCl_2_), With 5 mM citrate, supplemented with 0.5% ethanol or 20 mM potassium acetate. Each strain was grown at 37 °C with an initial OD_600_ of 0.1, with continuous shaking. The samples were collected periodically to analyze fluorescence and bacterial growth. Samples were diluted 10-fold in 0.9% NaCl, and fluorescence expression was measured using a Varioskan LUX Multimode Microplate Reader (3020-80145, ThermoFisher, Waltham, MA, USA) with excitation at 395 nm and emission at 509 nm. Samples for growth curve determination were collected simultaneously, diluted in 0.9% NaCl, and then assessed for changes in OD_600_ using the same instrument.

### 2.3. Purification of DJ41_2796 and Enzymatic Assay

Following the methodology described in previous research [18], the *DJ41_2796* gene was cloned into pQE80LK, which was derived by substituting the ampicillin resistance gene of pQE80L (Qiagen, Hilden, Germany) with kanamycin. A gene fragment of *DJ412796*, 1515 bp in length, was ligated with a pQE80LK cutting by *Bam*HI and *Pst*I. The resulting construct, pQE80LK-DJ41_2796, was then transformed into *E. coli* DH5α for protein expression and purification (Appendix A). A total of 100 mL of bacteria were cultured, with OD_600_ of 0.1 at initiation. Protein expression was induced by 0.5 mM isopropyl ß-D-1-thiogalactopyranoside (IPTG) after OD_600_ reached 0.6 for three hours. The cultures were then harvested and lysed using a high-pressure homogenizer CF1 (Constant Systems Ltd., Daventry, UK), and purification of DJ41_2796 from the supernatant was conducted at 4 °C using NGC Chromatography Systems (Bio-rad, Hercules, CA, USA). In this system, the supernatant was passed through a nickel-affinity column (Nuvia IMAC Ni-Charge, Bio-rad, Hercules, CA, USA). The histidine-tag fusion protein was purified by increasing the concentration of imidazole from 5 mM to 1 M in the buffer [17]. 

The purified protein was utilized for enzyme activity analysis in accordance with the experimental design of previous studies [23,24]. Each 1 mL reaction sample for testing the dose response of DJ41_2796 contained a final concentration of 100 mM phosphate buffer solution, 10 mM potassium acetate, 0.2 mM succinyl-CoA, 0~10 µM DJ41_2796, and 0.1 mM Ellman’s Reagent (5,5′-dithio-bis-(2-nitrobenzoic acid), DTNB). After the reaction between DTNB and CoA, TNB (extinction coefficient: 14.15 M^−1^cm^−1^) was produced [25]. The changes in OD_412_ were monitored every 30 s within a 1.5 min timeframe, using the Multiskan SkyHigh Microplate Spectrophotometer (ThermoFisher, Waltham, MA, USA). Enzyme activity was then determined according to the conversion rate of acetate into acetyl-CoA, in nanomoles per minute.

## 3. Results

### 3.1. Genes of EmaSR Regulons Were Upregulated in a Low-Concentration Ethanol Environment

Binding-box analysis was conducted on sequences of approximately 250 base pairs upstream of each gene, and a sequence of AAxCTTAxxxxTAxxxTTxxxx upstream of the EmaSR regulon was found to be highly conserved (Table 4). This same sequence was identified upstream of *emaS* and *emaR*, thereby suggesting that these genes might be regulated by EmaR. This study constructed plasmid TCSG to confirm that EmaSR regulons were indeed upregulated in low-ethanol conditions. TCSG enables the expression of *emaS* and *emaR*, along with *gfpuv* (Appendix A). Approximately 250 bp upstream of each EmaSR regulon were combined with the *gfpuv* of the TCSG reporter plasmid to construct derivative plasmids, and after confirming its successful construction and transformation using colony PCR (Appendix A), each clone was observed for fluorescence expression in LB medium containing 1% ethanol (Appendix A). However, differences in fluorescence expression among all strains, including the control group, did not exceed 6% of the average fluorescence value. There was no significant difference in the expression levels of each EmaSR regulon compared to the control group (Appendix A). Since gene expression in *E. coli* cultured in LB medium differs from the growth of *A. baumannii* in low-concentration ethanol conditions, we inferred that the TCSG reporter system in *E. coli* is unsuitable for analyzing EmaSR regulon expression; therefore, DNA fragments combining the regulon promoters with *gfpuv* were transferred from TCSG to the *E. coli* and *A. baumannii* shuttle vector pWH1266 (Appendix A) for further analysis. The successful establishment of each pWH1266 reporter strain was subsequently confirmed by the colony PCR (Appendix A).

During the analysis, it was observed that *A. baumannii* wild-type strains carrying reporter plasmids, such as *DJ41_566-571* (P*_571_*), *DJ41_2796* (P*_2796_*), *DJ41_3218* (P*_3218_*), and *DJ41_3568* (P*_3568_*), exhibited slightly higher fluorescence levels after 24 h compared to wild-type strains carrying an empty reporter plasmid (P*_gfp_*) under citrate conditions (Figure 2A). In addition, under 0.5% ethanol culture conditions, fluorescence expression levels of EmaSR regulons increased by four to six times over the control group (Figure 2C). Although growth curves leveled out after 24 h for all strains, statistically significant differences in fluorescence expression continued to be observed between different strains, indicating that the differences in fluorescence were unlikely to be influenced by growth fluctuation (Figure 2B,D). The fluorescence expression of P*_2796_* and P*_3218_* significantly increased under 0.5% ethanol culture conditions, respectively, reaching approximately 3-fold and 2-fold higher than the control (Figure 2C). By contrast, fluorescence expression levels for P*_571_* and P*_3568_* were not significantly different from P*_gfp_* at 48 h. These results suggest that *DJ41_2796* and *DJ41_3218* are regulated by EmaSR in ethanol-containing culture conditions. P*_emaR_* also exhibited significant differences in fluorescence expression compared to the control from 24 h onwards in 0.5% ethanol-containing cultures, but unexpectedly, there were no significant differences in the fluorescence expression for P*_emaS_* in comparison to the control. These results confirm that P*_emaR_*, P*_2796_*, and P*_3218_* are induced in the presence of ethanol (Figure 2C).

To further confirm the upregulation of EmaSR regulons, the expression levels of P*_2796_* and P*_3218_* were analyzed in wild-type and mutant strains (∆*emaS*, ∆*emaR*, and ∆*emaSR*), cultured under low-concentration ethanol conditions. Similar expression levels at 48 h were observed in the wild-type strain as previously noted (Figure 3 black and gray slash, Appendix A). However, in mutants ∆*emaS*, ∆*emaR*, and ∆*emaSR*, there was no significant elevation in fluorescence expression for P*_2796_*and P*_3218_* (Figure 3 black and gray slash, Appendix A). Although the wild-type strain grew approximately 4- to 6-fold higher than mutant strains initially, all strains reached similar growth levels after 48 h (Appendix A). This confirmed that at least by 48 h, the difference in expression levels of P*_2796_* and P*_3218_* between wild-type and mutant strains was not due to growth defects and demonstrated that the increase in fluorescence expression levels in low-concentration ethanol culture conditions was indeed regulated by EmaSR.

### 3.2. EmaSR Regulates DJ41_2796 and DJ41_3218 in Acetate Culture Conditions

In previous research, it was discovered that EmaSR may be involved not only in the regulation of ethanol metabolism, but also in acetate metabolism [18]. To confirm whether EmaSR regulates acetate metabolism via regulons, the fluorescence expression of EmaSR regulons was observed under low-concentration acetate culture conditions. After 24 h, fluorescence expression levels of EmaSR regulons were observed to be 3-fold higher than the P*_gfp_* control after 24 h, and the fluorescence expression for P*_3218_* reached 5-fold higher than P*_gfp_* (Figure 2E). A growth curve analysis confirmed no significant differences in the growth of each strain (Figure 2F). Comparable fluorescence expression levels of P*_emaR_* in acetate culture conditions to ethanol culture conditions were observed, with the fluorescence expression of P*_emaR_* significantly increased; however, there was a significant decrease in the fluorescence expression of P*_emaS_* compared to P*_gfp_* (Figure 2E), suggesting that *emaR* can be induced in the presence of acetate, but *emaS* cannot.

Fluorescence expression levels of P*_2796_* and P*_3568_* were observed in both wild-type and mutant *A. baumannii* strains, and as in ethanol-containing cultures, there was no significantly elevated fluorescence expression observed for P*_2796_* and P*_3568_* in all mutants. During the 24 h culture period, the fluorescence expression levels of P*_2796_* and P*_3218_* in the wild-type strain were both 4-fold higher than that of the mutant strains (Figure 4 black and gray slash, Appendix A). The growth curve analysis also confirmed that this difference was not due to growth differences (Figure 4 black and gray slash, Appendix A). This indicates that in low-concentration acetate culture conditions, EmaSR indeed regulates the expression of EmaSR regulons.

### 3.3. EmaR Enhances the Expression of P_emaS_ and P_emaR_ in Acetate

TCS are known to not only activate the regulation of downstream genes but also to control the expression of their own promoters. Previous results indicated significantly elevated expression of P*_emaR_* under low-concentration ethanol culture conditions [18], but here it was found that in a similar environment, P*_emaR_* was barely expressed in all three mutant strains (Figure 3 white, Appendix A). This difference was shown to be unaffected by growth (Appendix A) and confirmed that the expression of P*_emaR_* in low-concentration ethanol environments is also regulated by EmaSR. Similar results were observed under low-concentration acetate culture conditions (Figure 4 white, Appendix A), and the findings were not influenced by differences in growth (Appendix A).

By contrast, P*_emaS_* showed no significant increase in expressions in both ethanol and acetate, and even had a significantly lower expression in the acetate environment compared to the P*_gfp_* control. However, it was observed that in the ∆*emaR* and ∆*emaSR* mutants, fluorescence expression of P*_emaS_* was almost half that of wild type and ∆*emaS* in low concentrations of ethanol (Figure 3 dark gray, Appendix A). There was no significant variation in tye fluorescence expression levels for all strains in the acetate-containing medium (Figure 4 dark gray, Appendix A). The growth curve analysis confirmed that this result was not due to differences in growth rates (Appendix A). These results show that although the expression of P*_emaS_* was not induced by ethanol or acetate, its expression was still regulated by EmaR, suggesting that in strains with intact *emaR*, the expression of P*_emaS_* can still be enhanced. Additionally, although not significantly different, slightly higher expression levels of P*_emaS_* were observed in ∆*emaS* under both low ethanol and acetate conditions, as compared to the wild type. There may be other regulatory factors that suppress P*_emaS_* expression, of which are present in the wild-type strain but not ∆*emaS*.

### 3.4. DJ41_2796 Possesses Acetate: Succinyl-CoA Transferase Enzymatic Activity

Both transcriptome analysis and reporter assays confirmed the potentially significant role of DJ41_2796 in EmaSR-regulated ethanol and acetate metabolism, and a functional prediction using the amino acid sequence suggests that DJ41_2796 may have ASCT activity. The homologous genes of this enzyme have been proven to play a crucial role in the metabolic regulation of MxtR/ErdR [13]. The enzymatic activity of DJ41_2796 was subsequently assessed through interaction with potassium acetate and succinyl-CoA, to ascertain if acetyl-CoA and succinate would be produced. DJ41_2796 was expressed in *E. coli*, using the vector pQE80LK-DJ41_2796, and was purified through nickel affinity chromatography. Different concentrations of DJ41_2796 were tested, and as concentrations increased from 2 µM to 10 µM, the reaction rate also increased from 0.9611 U/g to 4.9823 U/g (Figure 5). This confirmed that DJ41_2796 possesses ASCT enzymatic activity to convert potassium acetate and succinyl-CoA into acetyl-CoA and succinate.

### 3.5. Kinetic Characterization of DJ41_2796

To better understand the enzymatic performance of DJ41_2796, reactions were tested under different pH and temperature conditions. It was observed that DJ41_2796 had the highest reaction rate (34.9 U/g) in a pH 8.0 phosphate buffer; however, in pH 8.0 Tris-Cl buffer, reaction rates were lower despite being at the same pH. Additionally, a decrease in reaction rates was noted in conditions with the pH exceeding 8.0 (Figure 6A). Subsequently, all enzymatic reactions with DJ41_2796 were carried out in the pH 8.0 phosphate buffer. Interestingly, it was noted that as the temperature increased from 4 °C to 55 °C, reaction rates also increased from 1.4 U/g to 21.7 U/g (Figure 6B), suggesting that DJ41_2796 may exhibit better reaction activity at higher temperatures. 

The enzyme kinetics of DJ41_2796 were assessed, using either potassium acetate or sodium acetate as substrates, in order to ascertain whether there was a difference in affinity for these two types of acetate. The K_M_ for potassium acetate was found to be approximately twice as high as that for sodium acetate, indicating that a higher concentration of potassium acetate is needed to reach half of V_max_. However, a further assessment of k_cat_/K_M_ revealed no significant difference in the reaction efficiency between these two substrates. This suggests that DJ41_2796 does not exhibit a significant difference in affinity for these two substrates (Table 5). The enzyme kinetics of DJ41_2796 with succinyl-CoA were also assessed, and it was observed that DJ41_2796 exhibited greater ASCT activity compared to a homologous protein in *Acetobacter aceti* (Table 5).

## 4. Discussion

### 4.1. Expression of Emas in Ethanol and Acetate May Be Inhibited by Other Regulatory Proteins

Previous research showed that EmaSR plays a role in *A. baumannii* ethanol metabolism [18]. In this study, the wild-type strain exhibited an increase in the expression levels of various promoters, including P*_emaR_* and the promoter of EmaSR regulons, in ethanol-containing environments. In acetate-containing environments, there was also an upward trend in expression levels, except for P*_emaS_*. Furthermore, it was observed that expression levels of P*_emaS_* in ∆*emaS* were higher than in the wild-type strain, regardless of whether the environment contained ethanol or acetate. In the absence of EmaS to transfer phosphate, EmaR may regulate downstream genes by receiving phosphate from other factors. There may also be other factors that act to inhibit EmaS in the wild-type strain, which could result in higher expression of P*_emaS_* in ∆*emaS* compared to the wild type. A similar scenario has been observed in *P. aeruginosa*, where it was found that the regulatory protein HapZ interacts with the REC domain of the sensor protein SagS when bound to cyclic di-GMP, thereby inhibiting phosphate transfer between SagS and downstream proteins [26]. Through SMART (Simple Modular Architecture Research Tool) analysis of its secondary structure, we found that the sensor protein EmaS also possesses a REC domain within its C-terminal structure, which can receive signals from sensor proteins and bind with DNA. Therefore, based on the functions predicted in the NCBI database, EmaS is also annotated as a hybrid sensor histidine kinase/response regulator, similar to SagS. From this, it can be inferred that EmaS may also have the ability to interact with other regulatory proteins, and in ethanol- or acetate-containing environments, we hypothesize that other factors may be involved in the suppression of signal transduction from EmaS to EmaR, and these may act via binding to the REC structural domain of EmaS.

### 4.2. DJ41_2796 Plays a Crucial Role in the Ethanol and Acetate Metabolism of A. baumannii ATCC19606

In a study on the toxicity of *A. pasteurianus* against acetate accumulation, it was confirmed that the expression levels of four proteins increased when exposed to high concentrations of acetate [27]. Overall, three of these proteins were annotated as TCS sensor proteins (two from the NtrB family and one from the SsrA family), and one was annotated as a BaeS and OmpR family regulatory protein [27]. It was also found that the expression of *ackA* (acetate kinase), *acs* (acetyl-CoA synthetase), *pta* (phosphate acetyltransferase), and *aarC* (ASCT) genes increased in high acetate environments [27]. Although it has not been confirmed that TCSs can directly regulate these genes, the results are indicative of the important role that TCSs and genes involved in acetate metabolism have with regard to *A. pasteurianus* resistance against acetic acid stress [27,28]. Interestingly, succinyl-CoA synthetase, which can convert succinyl-CoA into succinate and acts in a similar way to ASCT, is retained in many *Proteobacteria* species, as revealed by genome analysis; however, unlike *Proteobacteria*, most bacteria in the *Actinobacteria* and *Bacteroidetes* phyla predominantly possess only the gene for ASCT [29]. This suggests that ASCT-only bacteria may be using the ASCT enzyme to replace succinyl-CoA synthetase [29,30]. Genomic analysis of *A. aceti* has shown that the species lacks succinyl-CoA synthetase, and previous research showed that AarC6 in *A. aceti* may have replaced the function of succinyl-CoA synthetase [31]. DJ41_2796 and AarC in *A. aceti* exhibit the same enzymatic activity, and although the genome of *A. baumannii* ATCC 19606 retains a gene annotated as succinyl-CoA synthetase (*DJ41_3576*), transcriptional analysis of EmaSR confirmed that this gene is not regulated by EmaSR in low concentrations of ethanol [18]. While it is not clear whether DJ41_3576 functions as a succinyl-CoA synthetase, the results of this study affirm that DJ41_2796 may take on an important role in ethanol and acetate metabolism, replacing succinyl-CoA synthetase.

### 4.3. EmaSR May Influence the Pyruvate Metabolism by Regulating the Metabolism of Ethanol and Acetate

Previous research has demonstrated that in *Bacillus subtilis*, the enzyme MaeA is highly expressed only in the presence of malate, which is homologous to DJ41_3218. The study confirmed that *maeA* is regulated by YufL/YufM, which belongs to the CitA/CitB family of TCS and shares homologous proteins with CitA/CitB in *E. coli* and YdbF/YdbG in *K. pneumoniae* [32]. These TCSs have been previously shown to regulate the expression of genes involved in the citric acid (TCA) cycle. It was also confirmed that MaeA possesses enzymatic activity as a malate dehydrogenase, thereby substantiating the involvement of YufL/YufM in the regulation of the *B. subtilis* TCA cycle [32]. Subsequent studies further confirmed that LytS/LytT can increase the expression of the pyruvate transporter protein PftAB. However, when MaeA causes an accumulation of pyruvate within the bacterial cells, this inhibits LytS/LytT, thereby decreasing the expression of *pftAB* and preventing the loss of pyruvate from secondary carbon sources when bacteria deplete their preferred carbon sources, such as glucose and malate [33]. In this study, although the enzymatic function of *DJ41_3218* has not been confirmed, it is annotated to participate in pyruvate metabolism. The functional annotation of *DJ41_568-571* also indicates its involvement in pyruvate metabolism. These genes exhibit increased expression in *A. baumannii* when ethanol is used as a carbon source, and the results resemble the regulation observed in *B. subtilis* [33]. *A. baumannii* metabolizes ethanol and acetate as carbon sources to produce pyruvate, and when ethanol and acetate in the environment are depleted, this may upregulate pyruvate metabolism. A previous study showed that *B. subtilis* regulates pyruvate metabolism through two pairs of TCSs [32], and transcriptional analysis in our previous research [18] also found that more than ten genes annotated with regulatory protein functions are under the control of EmaSR [18]. Therefore, EmaSR may play an important regulatory role not just in ethanol and acetate metabolism but can also influence other metabolic processes.

### 4.4. EmaSR Regulation of Pyruvate Metabolism and the Relationship with Virulence

In this study, we observed that in ethanol-containing environments, the expression of P*_2796_* is higher than that of P*_3218_*. Conversely, in acetate-containing environments, the expression of P*_3218_* is significantly higher than that of P*_571_*, P*_2796_* and P*_3568_*. *DJ41_3218* is annotated as an NAD-dependent malic enzyme, primarily capable of converting malate into pyruvate. Previous research has also confirmed that this enzyme catalyzes the conversion of oxaloacetate to pyruvate in *E. coli* and *Lactobacillus arabinosus* [34]. On the other hand, within the EmaSR regulated gene cluster *DJ41_566-571*, the three genes *DJ41_568*, *DJ41_569*, and *DJ41_571* have been respectively annotated to encode dihydrolipoyl dehydrogenase, 2-oxo acid dehydrogenase subunit E2, and thiamine pyrophosphate (TPP)-dependent dehydrogenase E1, respectively. These three enzymes collectively work in the metabolic pathway to convert pyruvate into acetyl-CoA. In previous studies, it was found that *P. putida* KT2440 lost the ability to grow by using pyruvate as a single carbon source when *mxtR* and *erdR* were deleted [13]. Furthermore, YufL/YufM also participates in the regulation of pyruvate metabolism in *B. subtilis* [33]. Previous studies have already demonstrated that MxtR/ErdR and YufL/YufM can regulate pyruvate metabolism. Therefore, we speculate that EmaSR may regulate pyruvate metabolism via DJ41_3218.

Moreover, previous research has shown that the deletion of enzymes in the pyruvate metabolism pathway does not lead to a reduction in adhesins, but decreased bacterial virulence in *Streptococcus pneumoniae* [35]. In this study, it is suggested that this decrease may be indirectly caused by a lack of energy. Our previous study has also found that the loss of *emaS* and *emaR* led to a decrease in the virulence of *A. baumannii* when infecting *Galleria mellonella* larvae [18]. Taken together, it is possible that EmaSR may indirectly affect virulence by regulating pyruvate metabolism in *A. baumannii*.

## 5. Conclusions

In this study, we found that EmaSR increased the expression levels of *DJ41_566-571*, *DJ41_2796*, *DJ41_3218*, and *DJ41_3568*, as well as *emaR*, in ethanol- and acetate-containing environments. Additionally, biochemical functional analysis confirmed the role of DJ41_2796 in the carbon metabolic pathway. Results from the reporter assays showed that EmaSR increased the expression of DJ41_2796, leading to the conversion of more acetate to acetyl-CoA in the ethanol. These results furthered our understanding of EmaSR regulation in *A. baumannii*. When strains were cultured in a low-concentration ethanol or acetate environment, EmaSR was able to increase the expression of DJ41_2796, aiding in the metabolism of ethanol and acetate. This enables *A. baumannii* to increase the survival in ethanol and acetate while utilizing them as carbon sources for growth.

## Figures and Tables

**Figure 1 microorganisms-12-00331-f001:**
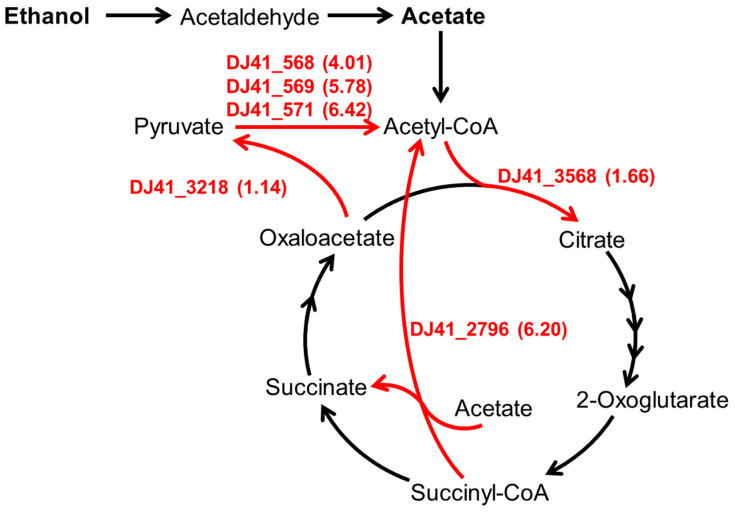
The carbon metabolism pathway regulated by EmaSR regulons. Based on gene function annotations from the National Center for Biotechnology Information (NCBI) database for *A. baumannii* ATCC 19606 and the metabolic pathways of the relevant Kyoto Encyclopedia of Genes and Genomes (KEGG) pathway. The numbers listed in parentheses beside the EmaSR regulons represent the log2 fold-change in gene expression between strains with and without *emaSR* in 0.5% ethanol, as derived from EmaSR transcriptome analysis.

**Figure 2 microorganisms-12-00331-f002:**
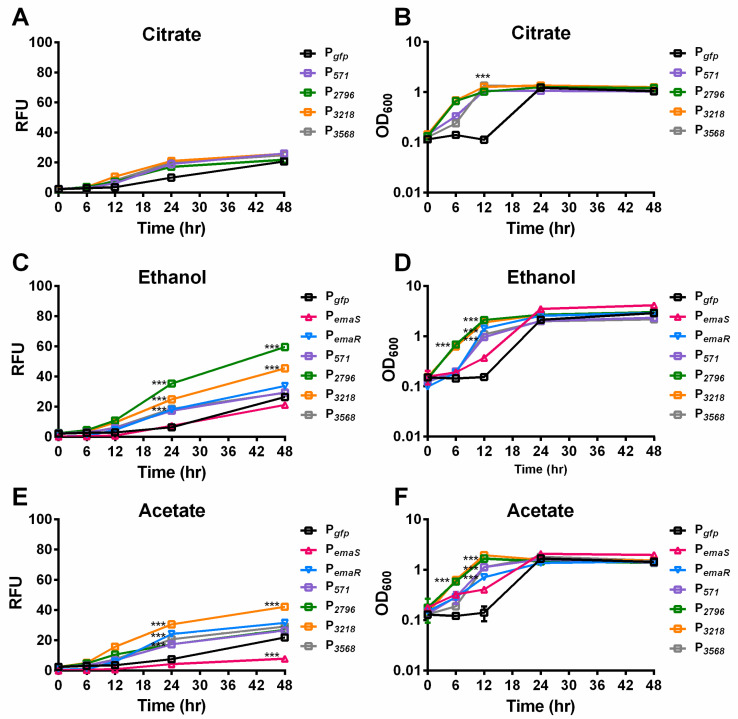
Mean fluorescence expression levels and growth of EmaSR regulon-transformed *A. baumannii* strains under 0.5% ethanol and 20 mM potassium acetate culture conditions. Wild-type strains of *A. baumannii* containing pWH1266G (black), pWH-P*_571_*G (purple), pWH-P*_2796_*G (green), pWH-P*_3218_*G (orange), pWH-P*_3568_*G (gray), pWH-P*_emaS_*G (red triangle), and pWH-P*_emaR_*G (blue inverted triangle) were grown in final concentrations of (**A**,**B**) 5 mM citrate; (**C**,**D**) 5 mM citrate and 0.5% ethanol; and (**E**,**F**) 5 mM citrate and 20 mM potassium acetate as carbon sources. Fluorescence expression (**A**,**C**,**E**) and growth curves (**B**,**D**,**F**) are shown. *** *p* < 0.0001, multiple *t*-test, derived from the comparison of strains with promoter and empty vector pWH1266G. RFU, relative fluorescence units. OD_600_, optical density. Data were collected from at least three replicates.

**Figure 3 microorganisms-12-00331-f003:**
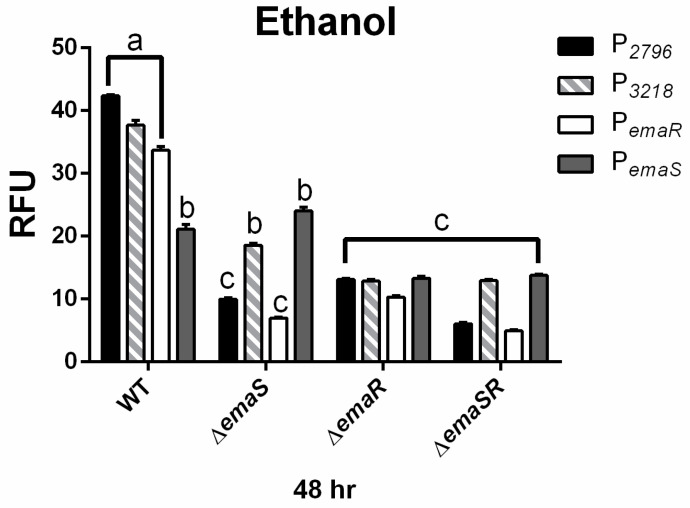
EmaSR regulon mean fluorescence expression levels in different *A. baumannii* strains cultured in low-concentration ethanol after 48 h. Changes in fluorescence levels at 48 h were observed in wild-type, ∆*emaS*, ∆*emaR*, and ∆*emaSR* strains of *A. baumannii*, each carrying the pWH-P*_2796_*G (black), pWH-P*_3218_*G (gray slash), pWH-P*_emaR_*G (white) or pWH-P*_emaS_*G (dark gray) reporter plasmid. Strains were cultured with 0.5% ethanol. The a, b, and c labels indicate the comparisons for which *p* < 0.0001, multiple *t*-test. RFU, relative fluorescence units. Data were derived from at least three independent experiments.

**Figure 4 microorganisms-12-00331-f004:**
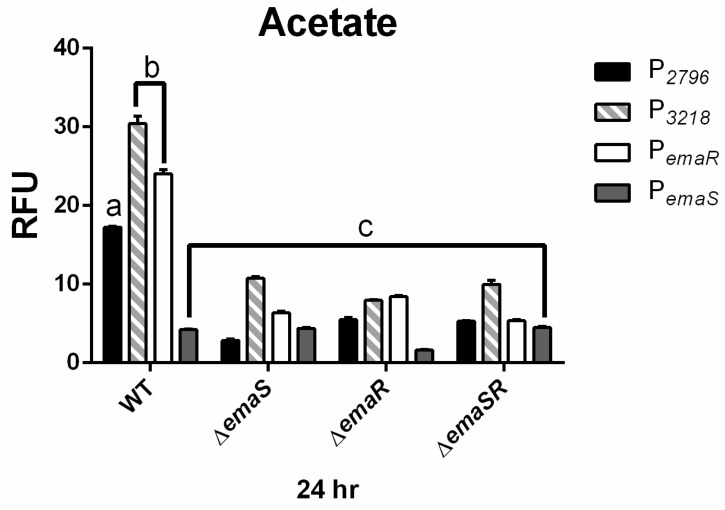
EmaSR regulon mean fluorescence expression levels in different *A. baumannii* strains cultured in low-concentration actetae after 24 h. Changes in fluorescence levels at 24 h were observed in wild-type, ∆*emaS*, ∆*emaR*, and ∆*emaSR* strains of *A. baumannii*, each carrying the pWH-P*_2796_*G (black), pWH-P*_3218_*G (gray slash), pWH-P*_emaR_*G (white) or pWH-P*_emaS_*G (dark gray) reporter plasmid. Strains were cultured with 20 mM acetate. The a, b, and c labels indicate the comparisons for which *p* < 0.0001, multiple *t*-test. RFU, relative fluorescence units. Data were derived from at least three independent experiments.

**Figure 5 microorganisms-12-00331-f005:**
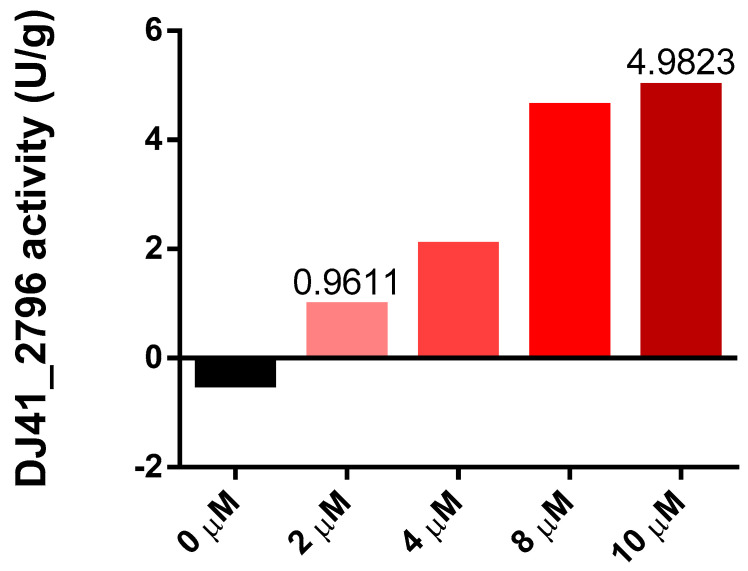
Enzymatic activity of DJ41_2796 at different protein concentrations. The y–axis represents the enzymatic activity of DJ41_2796 within one minute after subtracting the background value (U/g), and the x–axis represents the protein concentration of DJ41_2796. The reaction mixture contains a final concentration of 10 mM potassium acetate, 0.2 mM succinyl-CoA, 0.1 mM DTNB, and 0 to 10 µM DJ41_2796. The results were obtained from one experiment.

**Figure 6 microorganisms-12-00331-f006:**
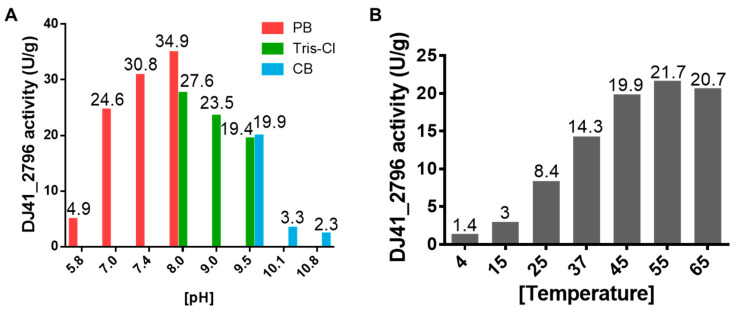
Assessing the optimal enzymatic activity of DJ41_2796. (**A**) Enzymatic activity of DJ41_2796 in buffer solutions with different pH values. Red bars represent phosphate buffer (PB), green bars represent Tris-HCl buffer (Tris-Cl), and blue bars represent carbonate–bicarbonate buffer (CB). The y–axis indicates the DJ41_2796 specific activity (U/g), and the numbers on top of each bar represent the activity of DJ41_2796 in that environment. (**B**) Enzymatic activity of DJ41_2796 under different temperature conditions, with the x–axis representing tested reaction temperatures (°C), and the y–axis indicating DJ41_2796 specific activity (U/g). Each number above the bars represents the enzymatic activity of DJ41_2796 in that environment. The results were obtained from one experiment.

**Table 1 microorganisms-12-00331-t001:** Bacterial strains used in this study.

No.		Description	References or Source
***Escherichia coli* (*E. coli*)**			
1	DH5α	F^−^, *supE44*, *hsdR17*, *reA1*, *gyrA96*, *endA1*, *thi-1*, *relA1*, *delR*, λ^−^	ATCC 53868
2	DH5α/pQE80LK-DJ41_2796	DH5α contain pQE80LK-DJ41_2796, Km^r^	This study
3	DH5α/TCSG	DH5α contain TCSG, Ap^r^	This study
4	DH5α/TCS-P*_emaS_*G	DH5α contain TCS-P*_emaS_*G, Ap^r^	This study
5	DH5α/TCS-P*_emaR_*G	DH5α contain TCS-P*_emaR_*G, Ap^r^	This study
6	DH5α/TCS-P*_571_*G	DH5α contain TCS-P*_571_*G, Ap^r^	This study
7	DH5α/TCS-P*_2796_*G	DH5α contain TCS-P*_2796_*G, Ap^r^	This study
8	DH5α/TCS-P*_3218_*G	DH5α contain TCS-P*_3218_*G, Ap^r^	This study
9	DH5α/TCS-P*_3568_*G	DH5α contain TCS-P*_3568_*G, Ap^r^	This study
10	DH5α/pWH1266	DH5α contain pWH1266, Ap^r^; Tet^r^	[19]
11	DH5α/pWH1266G	DH5α contain pWH1266G, Tet^r^	This study
12	DH5α/pWH1266-P*_emaS_*G	DH5α contain pWH1266-P*_emaS_*G, Tet^r^	This study
13	DH5α/pWH1266-P*_emaR_*G	DH5α contain pWH1266-P*_emaR_*G, Tet^r^	This study
14	DH5α/pWH1266-P*_571_*G	DH5α contain pWH1266-P*_571_*G, Tet^r^	This study
15	DH5α/pWH1266-P*_2796_*G	DH5α contain pWH1266-P*_2796_*G, Tet^r^	This study
16	DH5α/pWH1266-P*_3218_*G	DH5α contain pWH1266-P*_3218_*G, Tet^r^	This study
17	DH5α/pWH1266-P*_3568_*G	DH5α contain pWH1266-P*_3568_*G, Tet^r^	This study
***Acinetobacter baumannii* (*A. baumannii*)**			
18	ATCC 19606	Ap^r^, clinical isolate, wild type	[20]
19	ATCC 19606/pWH1266G	ATCC 19606 contain pWH1266G, Ap^r^, Tet^r^	This study
20	ATCC 19606/pWH1266-P*_emaS_*G	ATCC 19606 contain pWH1266-P*_emaS_*G, Ap^r^, Tet^r^	This study
21	ATCC 19606/pWH1266-P*_emaR_*G	ATCC 19606 contain pWH1266-P*_emaR_*G, Ap^r^, Tet^r^	This study
22	ATCC 19606/pWH1266-P*_2796_*G	ATCC 19606 contain pWH1266-P2796G, Apr, Tet^r^	This study
23	ATCC 19606/pWH1266-P*_3218_*G	ATCC 19606 contain pWH1266-P*_3218_*G, Ap^r^, Tet^r^	This study
24	ATCC 19606/pWH1266-P*_3568_*G	ATCC 19606 contain pWH1266-P*_3568_*G, Ap^r^, Tet^r^	This study
25	∆*emaS*	ATCC 19606, ∆*emaS*, Ap^r^	[18]
26	∆*emaS*/pWH1266G	∆*emaS* contain pWH1266G, Ap^r^, Tet^r^	This study
27	∆*emaS*/pWH1266-P*_emaS_*G	∆*emaS* contain pWH1266-P*_emaS_*G, Ap^r^, Tet^r^	This study
28	∆*emaS*/pWH1266-P*_emaR_*G	∆*emaS* contain pWH1266-P*_emaR_*G, Ap^r^, Tet^r^	This study
29	∆*emaS*/pWH1266-P*_571_*G	∆*emaS* contain pWH1266-P*_571_*G, Ap^r^, Tet^r^	This study
30	∆*emaS*/pWH1266-P*_2796_*G	∆*emaS* contain pWH1266-P*_2796_*G, Ap^r^, Tet^r^	This study
31	∆*emaS*/pWH1266-P*_3218_*G	∆*emaS* contain pWH1266-P*_3218_*G, Ap^r^, Tet^r^	This study
32	∆*emaS*/pWH1266-P*_3568_*G	∆*emaS* contain pWH1266-P*_3568_*G, Ap^r^, Tet^r^	This study
33	∆*emaR*	ATCC 19606, ∆*emaR*, Ap^r^	[18]
34	∆*emaR*/pWH1266G	∆*emaR* contain pWH1266G, Ap^r^, Tet^r^	This study
35	∆*emaR*/pWH1266-P*_emaS_*G	∆*emaR* contain pWH1266-P*_emaS_*G, Ap^r^, Tet^r^	This study
36	∆*emaR*/pWH1266-P*_emaR_*G	∆*emaR* contain pWH1266-P*_emaR_*G, Ap^r^, Tet^r^	This study
37	∆*emaR*/pWH1266-P*_571_*G	∆*emaR* contain pWH1266-P*_571_*G, Ap^r^, Tet^r^	This study
38	∆*emaR*/pWH1266-P*_2796_*G	∆*emaR* contain pWH1266-P*_2796_*G, Ap^r^, Tet^r^	This study
39	∆*emaR*/pWH1266-P*_3218_*G	∆*emaR* contain pWH1266-P*_3218_*G, Ap^r^, Tet^r^	This study
40	∆*emaR*/pWH1266-P*_3568_*G	∆*emaS* contain pWH1266-P*_3568_*G, Ap^r^, Tet^r^	This study
41	∆*emaSR*	ATCC 19606, ∆*emaSR*, Ap^r^	[18]
42	∆*emaSR*/pWH1266G	∆*emaSR* contain pWH1266G, Ap^r^, Tet^r^	This study
43	∆*emaSR*/pWH1266-P*_emaS_*G	∆*emaSR* contain pWH1266-P*_emaS_*G, Ap^r^, Tet^r^	This study
44	∆*emaSR*/pWH1266-P*_emaR_*G	∆*emaSR* contain pWH1266-P*_emaR_*G, Ap^r^, Tet^r^	This study
45	∆*emaSR*/pWH1266-P*_571_*G	∆*emaSR* contain pWH1266-P*_571_*G, Ap^r^, Tet^r^	This study
46	∆*emaSR*/pWH1266-P*_2796_*G	∆*emaSR* contain pWH1266-P*_2796_*G, Ap^r^, Tet^r^	This study
47	∆*emaSR*/pWH1266-P*_3218_*G	∆*emaSR* contain pWH1266-P*_3218_*G, Ap^r^, Tet^r^	This study
48	emaSR/pWH1266-P*_3568_*G	∆*emaSR* contain pWH1266-P*_3568_*G, Ap^r^, Tet^r^	This study

**Table 2 microorganisms-12-00331-t002:** Plasmids used in this study.

No		Description	References or Source
1	pQE80LK	expression vector, pUC ori; Km^r^	This study
2	pQE80LK-DJ41_2796	pQE80LK/*DJ41_2796*; Km^r^	This study
3	TCSG	pGFPuv/*emaS*; *emaR*; *gfpuv*; Ap^r^	This study
4	TCS-P*_emaS_*G	pGFPuv/*emaS*; *emaR*; P*_emaS_*-*gfpuv*; Ap^r^	This study
5	TCS-P*_emaR_*G	pGFPuv/*emaS*; *emaR*; P*_emaR_*-*gfpuv*; Ap^r^	This study
6	TCS-P*_571_*G	pGFPuv/*emaS*; *emaR*; P*_571_*-*gfpuv*; Ap^r^	This study
7	TCS-P*_2796_*G	pGFPuv/*emaS*; *emaR*; P*_2796_*-*gfpuv*; Ap^r^	This study
8	TCS-P*_3218_*G	pGFPuv/*emaS*; *emaR*; P*_3218_*-*gfpuv*; Ap^r^	This study
9	TCS-P*_3568_*G	pGFPuv/*emaS*; *emaR*; P*_3568_*-*gfpuv*; Ap^r^	This study
10	pWH1266	*E. coli*–*A. baumannii* shuttle vector; Ap^r^; Tet^r^	[19]
11	pWH1266G	pWH1266G/*gfpuv*; Tet^r^	This study
12	pWH1266-P*_emaS_*G	pWH1266G/P*_emaS_*-*gfpuv*; Tet^r^	This study
13	pWH1266-P*_emaR_*G	pWH1266G/P*_emaR_*-*gfpuv*; Tet^r^	This study
14	pWH1266-P*_571_*G	pWH1266G/P*_571_*-*gfpuv*; Tet^r^	This study
15	pWH1266-P*_2796_*G	pWH1266G/P*_2796_*-*gfpuv*; Tet^r^	This study
16	pWH1266-P*_3218_*G	pWH1266G/P*_3218_*-*gfpuv*; Tet^r^	This study
17	pWH1266-P*_3568_*G	pWH1266G/P*_3568_*-*gfpuv*; Tet^r^	This study

**Table 3 microorganisms-12-00331-t003:** Primers used in this study.

No.		Sequences (5′-3′)	References or Source
1	pQE80LK-DJ41_2796-eF	AAGGATCCATGTCTTTAAGTCGTATT	This study
2	pQE80LK-DJ41_2796-eR	AACTGCAGTTAAGCTGATTTTGCAAC	This study
3	TCSG-P*_emaS_*-gF	GGCTGCAGTAACCAAACTCCTTACATAG	This study
4	TCSG-P*_emaR_*-gF	GGCTGCAGCCCGAATAGTTGATTTTTAT	This study
5	TCSG-P*_571_*-gF	GGCTGCAGTTTGTTTTACAAGTATATGA	This study
6	TCSG-P*_2796_*-gF	GGCTGCAGGCGCTATTTTAAACCTCAAA	This study
7	TCSG-P*_3218_*-gF	GGCTGCAGGTCAGATATAGAGTTGAGAA	This study
8	TCSG-P*_3568_*-gF	GGCTGCAGTCCCGATGCAGTGATTTTAC	This study
9	pWH-P*_gfp_*-F	ACGTTGTTGCCATTGCTGCAAAAAATCTAATGCATGCCTGC	This study
10	TCSG-P*_emaS_*-gR	CCTCTAGAATGACCTACATAAGTGAAAC	This study
11	TCSG-P*_emaR_*-gR	CCTCTAGAAAGCGATTAAAGTAATCTTG	This study
12	TCSG-P*_571_*-gR	CCTCTAGAATCCTTTTCCTTTATTATCT	This study
13	TCSG-P*_2796_*-gR	CCTCTAGATGGACATCCTCAATATTGTC	This study
14	TCSG-P*_3218_*-gR	CCTCTAGATGCCTATCTCATTTTCCAGC	This study
15	TCSG-P*_3568_*-gR	CCTCTAGATCAATAGATCTCCTGTCCTG	This study
16	pWH-P*_gfp_*-reR	GATAAGCTGTCAAACATGAGCATTATTTGTAGAGCTCATCCA	This study
17	GFP-34R	TTCACCCTCTCCACTGACAGA	This study
18	TcR	GATGCGTCCGGCGTAGAG	This study
19	P-Ab-ITSB	AGAGCACTGTGCACTTAAG	[21]
20	P-Ab-ITSF	CATTATCACGGTAATTAGTG	[21]

**Table 4 microorganisms-12-00331-t004:** EmaR binding boxes on EmaSR regulons.

Name	Ratio	Strand	Position *	Sequence (5′-3′) ^†^
*DJ41_566-571*	2.85~6.42	**+**	−109 to −130	**AA**A**CTTA**TTTAA**A**ACT**TT**TTAG
*DJ41_3218*	1.14	**+**	−64 to −85	**A**GT**CTTA**AGCT**TA**CGCA**T**ACAA
*DJ41_3568*	1.66	**+**	−74 to −95	**AA**T**CTTA**TAGCA**A**ATT**TT**GACA
*DJ41_2796*	6.21	**-**	−105 to −126	TAAA**AA**TCA**TA**AAAA**TAAG**T**T**A
*emaR*	2.94	**-**	−14 to −35	GGCC**A**TACT**TA**TGCTC**AAG**A**TT**
*emaS*	2.30	**-**	−18 to −39	GGGTT**A**TGA**TA**GGCA**TAAG**G**TT**

* Start codon (Met) set as 0. ^†^ Underlined base pairs perfectly match AAxCTTAxxxxTAxxxTTxxxx, while **Bold** base pairs are identical to the highly conserved sequence that was found in every sequence analyzed here; analysis was conducted to 250 bp upstream of the start codon.

**Table 5 microorganisms-12-00331-t005:** Enzyme kinetics of DJ41_2796 for different substrates in pH 8.0 phosphate buffer.

	Potassium Phosphate Buffer, pH = 8.0		
	Potassium Acetate *	Sodium Acetate *	Succinyl-CoA *
V_max_ (nmole·min^−1^)	41.55	33.51	34.97
K_M_ (mM)	39	14.74	0.3418
k_cat_ (s^−1^)	8.31	6.702	6.994
k_cat_/K_M_ (mM^−1^s^−1^)	0.2131	0.4547	20.4623

* The results were obtained from one experiment.

## Data Availability

The data presented in this study are available upon reasonable request from the corresponding author.

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
