# Peer review of "Gene Expression of Ethanol and Acetate Metabolic Pathways in the Acinetobacter baumannii EmaSR Regulon"

_microorganisms, 2024, doi:10.3390/microorganisms12020331_

Round 1
Reviewer 1 Report
Comments and Suggestions for Authors
The manuscript by Huang et al explores the impact of ethanol and acetate on the transcriptional induction of an array of genes via promoter fusion reporter assays. While these genes are known to be differentially regulated in response to ethanol metabolism, the direct impact of these carbon sources has not been shown. While the manuscript does highlight some interesting observations, the authors should address the following concerns prior to publication.
Figures and Tables
- Figure 1 is not introduced in the text of the manuscript and the figures are not introduced in sequential order (i.e. currently supplementary figure 5 is the first figure to be introduced)
- All figure legends – please indicate what the data points represent (ie mean/median), how many biological/technical replicates are represented for each timepoint and include error bars.
- As a general recommendation please avoid the use of red and green on the same figure.
- I’m not sure why Table 1 is divided into 3 separate sections. Furthermore, either as part of the strain descriptions or the methods would benefit from being expanded to provide more detailed information (i.e. for the promoter fusion constructs how large a fragment was cloned in and using what restriction sites)
- Personally, I’m not sure that figure 2 adds anything to the manuscript, could be moved to supplementary and the discussion of the data significantly reduced? Given all the constructs are investigated thoroughly in the subsequent figures, the analysis seems somewhat obsolete.
- Could figures 4,5,6,7 be combined in some way? Or at least reduced to plot RFU relative to OD (reducing the numbers of graphs by half, but also making it easier for the reader to compare between samples). I understand these look at different promoter regions in the different mutant backgrounds but it’s very repetitious – Could the full figures be moved to supplementary and a single timepoint for all data be presented on graph in the main manuscript? This would allow the reader to more easily compare between strains and promoters across the conditions.
- Please include ± value for each calculated concentration to highlight the range of error/variation in Table 5. Otherwise if these values are derived from a single replicate this needs to be made clear.
Methods
- Line 100, is ‘TCSG’ an abbreviation? If so, please define.
- Line 101, please provide more details regarding what the significance/difference of ‘gfpuv’ is compared to regular ‘gfp’
- Line 106, please provide a reference or more information regarding the composition of the M9 media used, as numerous variations of this media exist.
- Line 120, please provide more information concerning the ‘harvesting of the cultures’ – i.e. what growth phase/OD were the cells at.
- Line 124, the nickel affinity column purification – please provide more details, i.e. buffer composition, conditions, etc?
Line 127, please provide details of what the buffer solution was?
Results
- Line 138 – please rephrase to be clearer regarding what constitutes the upstream region, i.e. from the start codon or transcriptional start site? Furthermore, if this was from the start codon, how did the authors confirm that this included the transcriptional start site and any upstream regulatory region.
- Line 139 – please define what is meant by ‘highly’
Reviewer 2 Report
Comments and Suggestions for Authors
In the manuscript "Gene Expression of EmaSR Regulon" the Authors research the EmaSR regulon, its activation and action when A. baumannii encounter ethanol and acetate. The manuscript is quite well written, however I have some concerns about the Authors' interpretation of their results that I would like the Authors to address.
Major remarks:
1. The title of the manuscript would be more informative, if it contained more specific information - A. baumannii, metabolites and/or pathways used, etc.
2. How conserved are the investigated genes in A. baumannii? Their prevalence in the genus and/or species could indicate their significance for this opportunistic pathogen.
3. In the Introduction, more information about the importance of A. baumannii to resist ethanol treatment would make the topic discussed more impactful.
Several other questions (Q4-6) concern the fitness of the bacteria analysed - it should be clearly stated in the text, if the fitness of the strains is the same or not:
4. Starting from Fig. 3 (also, figures 4 and 5) the Authors claim “Comparable growth curves for all strains after 24 hours indicated that the observed fluorescence variation was not caused by growth fluctuation”; however, even if the growth curves even out at 24 h, there is a significant growth difference in the first 12 h, and no information between hours 12 and 24. The Authors should clearly present why they are calling the growth curves comparable.
5. If the growth is different due to the genetic background (the deletions, plasmids present) perhaps it would be more informative to consider evaluating RFU/OD ratio?
6. In the growth experiments, succinate is used alone as control, or together with ethanol/acetate. In the latter case, there concentration of total carbon sources is higher for the growing bacteria. Have the Authors considered this effect when analysing their data?
7. I disagree with the conclusion “This enables A. baumannii to reduce the damage caused by ethanol and acetate while utilizing them as carbon sources for growth” – the Authors have not showed the reduction of damage in their results.
Minor remarks:
1. The abbreviations in the text should follow the same pattern. Was the Adh abbreviation not in capital letters by design?
2. Fig. 1 is not referenced in the text
3. In the Methods, it is first described that all bacteria were grown in LB (2.1), however, it seems that major part of experiments was done in M9. This should be clearly defined.
4. Unclear origin of gfpuv
5. In the figures, when a statistically significant difference is indicated by asterisks, it is unclear which points are taken for comparison
6. As I understood, the purification of the protein was performed using affinity chromatography; a little more information about the tag-fusion should be given in the text, as the fusion can influence the activity of the protein
Round 2
Reviewer 2 Report
Comments and Suggestions for Authors
I would like to thank the Authors for thoroughly addressing the questions and issues raised. I think the manuscript has signifficantly improved.